# Barriers and limitations to the development of a telemental health service for workers in Peru- A user-centered approach

John Astete Cornejo[1], Liliana Cruz-Ausejo[1]*, Jimmy Cainamarks Alejandro[1], Miguel Angel Burgos-Flores[1], Juan Ambrosio-Melgarejo[1], Jaime Rosales Rimache[1], Sebastián García Cruz[2]

**1** Centro Nacional de Salud Ocupacional y Protección del Ambiente para la Salud, Instituto Nacional de Salud, Lima, Peru, **2** Facultad de Ciencias Sociales, Pontificia Universidad Católica del Perú, Lima, Peru

\* rcruzausejo@gmail.com

## Abstract

### Introduction

Over the past decade, the surge in digital healthcare services has transformed traditional healthcare, requiring multidisciplinary engagement to adapt to the digital realm. The rise of telehealth services, particularly amid COVID-19, has been widely embraced globally, but its implementation in Peru faces unique challenges, including infrastructure issues and economic constraints. Therefore, this research aims to identify the barriers and limitations in developing a telemental health service for screening, evaluation, and timely referral of vulnerable occupational groups.

### Materials and methods

A qualitative study was undertaken. We adopted a phenomenological approach, utilizing semi-structured interviews with vulnerable occupational groups and decision-makers. We conducted 23 interviews: 5 providers of telemental health services, 5 teachers users, 5 police officers users and 5 health professionals of telemental health services, and 3 decision-makers involved in telemental health regulation in Peru.; exploring experiences, barriers, and facilitators related to mental telemental health. The interviews were recorded and transcribed verbatim, furthermore, a thematic analysis was done to identify key themes.

### Results

The research identified barriers and limitations to developing a telemental health services screening service based on the experiences of workers, some of them were related to user dissatisfaction, emphasizing the need for personalized solutions beyond technical aspects. Scheduling issues call for flexibility and improved communication. Healthcare professionals' varied experiences highlight the necessity for targeted training, while successful telemental health services integration demands addressing resource limitations through a comprehensive approach.

**Data availability statement:** The full transcripts of the interviews conducted with the participants, along with the corresponding Atlas.TI code file, are provided as Supporting Information. These data are openly accessible at the following link: https://doi.org/10.6084/m9.figshare.28147406.v1.

**Funding:** This work was funded by the Agencia Española de Cooperación Internacional para el Desarrollo (AECID, in Spanish) by Programa Nacional de Investigación Científica y Estudios Avanzados- CONCYTEC- Perú, under contract: N° 076-2021-PROCIENCIA.

**Competing interests:** The authors have declared that no competing interests exist.

## Conclusion

The study advocates for a holistic, user-centred paradigm in telemental health services implementation, addressing both technological aspects and human and systemic elements. Multifaceted challenges inherent in telemental health, particularly in Peru, emphasize the need for strategic interventions by stakeholders. The study calls for a policy shift towards enhancing telemental health equity through system-level changes and eliminating structural barriers for marginalized populations.

## Introduction

In the last decade, the surge in Telehealth has prompted a vital transformation for traditional healthcare businesses. This shift requires engagement from multidisciplinary teams and stakeholders to adapt to the digital world, ensuring optimal patient care. The thriving telehealth method, marked by recent activity and substantial investment, is challenging global healthcare services [1].

Globally, telehealth has emerged as a recurring strategy employed by governments and their health systems to ensure uninterrupted care provision amidst the COVID-19 pandemic [2]. In the evolving healthcare landscape, practitioners in low and middle-income countries (LMICs) have widely accepted these services [3]. In the specific context of Peru, as part of the Latin American region, the COVID-19 pandemic stimulated the rapid adoption of telehealth. However, its implementation in Peru faces unique challenges, including infrastructure issues, lack of electricity in some areas, limited access to an internet connection, and economic constraints hindering widespread access to electronic devices necessary for telehealth services [4].

In the field of Telehealth, Telemental Health refers to the use of telecommunications or videoconferencing technologies to deliver mental health services and interventions, including promotion, prevention programs, therapy, evaluations, and medication management [5,6]. This modality allows for remote care and has proven effective for various conditions, such as PTSD, depression, and anxiety. It is not globally regulated, even in contexts such as Peru, but has gained widespread acceptance in the context of the COVID-19 pandemic [7–9].

In Peru, the advent of telehealth predates the COVID-19 lockdown, with various initiatives started by non-governmental and educational entities, involving multiple projects in rural communities [4]. Presently, the Ministry of Health (MINSA) holds exclusive responsibility for the implementation of telehealth services in the country, encompassing both synchronous and asynchronous modalities [10]. Synchronous interactions involve real-time engagement between healthcare providers and patients, while the asynchronous approach entails deferred communication, with transmitted information subsequently reviewed by healthcare professionals [10]. It is noteworthy, however, that the Ministry of Health faces certain limitations as the sole regulatory entity for telemental health in Peru [11].

Concerning the mental health status of the Peruvian population, empirical evidence indicates that a substantial proportion, ranging from 28% to 41.3%, experienced mental health challenges during the pandemic lockdown [12,13]. Consequently, the imperative to address concerns related to the prevalence of adverse mental health outcomes in the Peruvian population has driven the Peruvian health system to confront novel challenges, exemplified by the integration of e-screening methodologies designed to discern and potentially address mental health issues. Notably, the global experience underscores that, despite the widespread activation of various e-screening initiatives in routine clinical practice throughout the COVID-19 era, comprehensive studies evaluating their appropriateness, efficiency, and efficacy remain remarkably scarce [14]. In the Peruvian context, telemental health assumes a main role in bridging this existing gap.

Peruvian regulatory framework defines telehealth as the provision of health services delivered through communication and information technologies, thereby organizing and advocating for the widespread use of this service [15]. Notably, during the pandemic, telehealth services saw a general increase in both supply and patient demand, so much so that the Ministry of Health reported a dramatic increase in the daily use of telehealth services provided by Infosalud in the first 6 months of the pandemic [16], and further noted that approximately 70% of telehealth corresponded to mental health issues [17].

However, these telehealth services focus on treating patients, leaving a gap concerning early identification through screening and initial management to prevent disease progression. Our research aims to identify the barriers and limitations for the future development of a telemental health service for screening, evaluation and timely referral in mental health based on the experiences of workers who are susceptible to these issues and also from the health workers who provide the service and the decision-makers. These vulnerable occupational groups have been particularly affected, as is the case with healthcare workers, police personnel, and educators who have faced a substantial increase in workload, direct exposure to COVID-19, shortages of personal protective equipment, abuse, and discrimination, among other challenges [18–25].

## Materials and methods

### Study design and methods

This qualitative study employed a phenomenological approach, immersing itself in data to interpret generated meanings that reflect participants' subjective realities [26]. The verbatim transcription of interviews played a role in ensuring effective interpretation [26]. Utilizing Interpretative Phenomenological Analysis (IPA) methods, the research conducted semi-structured interviews to explore the experiences of service users, health workers, and decision-makers from urban areas. Designed with specific questions, these interviews aimed to facilitate a dialogue that would uncover new aspects aligned with the research objectives [27]. Spanning durations of 10–35 minutes, the interviews aimed to understand perceptions, needs, barriers, and facilitators related to implementing telemental health services. The interviews were conducted by trained qualitative researchers. The analysis was carried out using Atlas.ti 8, including coding based on textual quotes, information structuring, and the consolidation of the telehealth platform evaluation across the diverse study participants, occurring after the familiarization with transcribed interviews.

### Data collection methods

The base questions of the semi-structured interview, as approved in the research protocol, are attached as a Supporting information file. These questions introduced the topic, they allowed the interviewer to ask follow-up questions and explore certain criteria in greater depth. The interviews were conducted by two professionals in psychology and one in anthropology, all with training and prior experience in qualitative studies. None of the interviewers had prior experience in telehealth or belonged to an entity related to the regulation or provision of telehealth services before the study. Therefore, the conduct of the interviews was not influenced by preconceptions or the interviewers' interests.

### Policemen, health professionals and teachers

We conducted a total of 15 interviews with various user groups, including health professionals, teachers, and police officers. The inclusion criteria comprised individuals who scored high on depressive or anxious symptoms, determined by the PHQ-2 and GAD-2 tools. This

assessment was integrated into the occupational activities conducted by CENSOPAS-INS (*Centro Nacional de Salud Ocupacional y Protección del Ambiente para la Salud- Instituto Nacional de Salud*, in Spanish).

Police officers were invited by DIRSAPOL (*Dirección de la sanidad policia*l, in Spanish; the police health directorate) and were selected using intentional non-probabilistic sampling. Meanwhile, teachers were invited through UGEL 3 (Unidad de Gestion Educativa local, in Spanish; a local educational entity in Perú).

### Telehealth professionals

In addition, we conducted five interviews with healthcare professionals engaged in teleconsultations, primarily psychologists or physicians providing mental health care within Community Mental Health Centers in Lima or other MINSA health facilities. Specifically, these professionals were sourced from the Community Mental Health Centers of the Ministry of Health (MINSA), including San Gabriel Alto, La Perla, and Carabayllo.

### Decision-makers

We have conducted three interviews with decision-makers, defined as public officials whose responsibilities are associated with administrative, managerial, or decision-making aspects of telemental health. These individuals may represent institutions such as the Directorate of Mental Health (DSAME, in Spanish) within MINSA, the General Directorate of Telehealth, Emergencies, and Referrals (DIGTEL, in Spanish) of MINSA, and EsSalud's National Telehealth Center (CENATE, in Spanish). Decision-makers were invited to participate in the interviews through formal letters sent by the Directorate of CENSOPAS-INS.

### Ethical statement and informed consent

Informed consent was obtained before conducting each of the 23 interviews. Participants received an overview of the study, along with the informed consent document. Upon orally expressing their agreement to participate, audiovisual or voice recording, as applicable, was initiated. Additionally, the explanation and informational sheet explicitly communicated participants' autonomy to withdraw from the study at any point if they wished to do so. The study protocol was approved by the Research and Ethic Institutional Committee of the INS (Instituto Nacional de Salud - Peru) under Resolution: RD N°206–2023-OGITT/INS, code OC-048–2022. Data collection occurred from June 23 to September 9, 2023.

### Data analysis

In the analysis of our findings, we have organized them into themes and subthemes derived from the data collected through semi-structured interviews. The interviews were analyzed in Atlas.ti using a pre-established codebook. An initial inductive coding of the transcribed data was conducted, allowing for the identification of emerging categories. Subsequently, an axial coding process was structured to group categories into main themes, following a triangulation strategy among researchers to ensure the validity of the analysis. An iterative review of the codes was carried out to ensure that interpretations remained consistent with the participants' narratives. The barriers and facilitators were identified using a holistic and user-centered approach.

## Results

Coding our findings was a dynamic process, which ultimately refined our coding framework and focused our attention on the four key themes associated with the mental health telecare

and screening service. The main themes and sub-themes that emerged from this process are detailed in Table 1.

## Telemental health does not satisfy users

The dissatisfaction expressed by users with telemental health services highlights a significant limitation of remote medical care. One user noted their discomfort with the absence of the physical presence of a doctor during teleconsultations, emphasizing the impersonal nature of the interaction. The sentiment expressed resonates with the challenges of providing holistic care through telehealth, where the inability to observe non-verbal cues and physical conditions may impact the quality of mental health assessments.

> "I found it unsatisfactory that, as I mentioned, the doctor isn't physically present to assess one's mental state. They simply provide a prescription, listen to you, and based on that, recommend a list of medications for you to pick up. It feels like a distant form of care."
>
> -D9: High-school teacher-

Furthermore, the desire for a comprehensive medical record points to a critical need for better integration of digital platforms in telemental health. The absence of a centralized and accessible medical history for users creates a gap in continuity of care. This limitation becomes apparent in scenarios where ongoing follow-up is required, and the lack of a consolidated medical record hinders the effectiveness of subsequent interactions.

> "Subsequently, they provided ongoing follow-up. A doctor called me the next day, even in the evening, and continued to do so until the third and fourth calls. However, when I reached out again, what I hoped for was a comprehensive medical record in place. This way, when a person calls again, they already have access to their history, and the healthcare providers are aware of the prescribed treatments."
>
> -D12: Elementary school Teacher-

These user experiences underscore the importance of addressing not only the technical aspects of telemental health but also the broader aspects of user satisfaction and continuity of care. Integrating solutions that provide a more personalized and comprehensive healthcare experience could enhance the overall effectiveness and user satisfaction with telemental health services.

**Table 1. The main themes and sub-themes identified.**

| Topics | Subtopics |
|---|---|
| **Receiver of the services** <br> 1. Telemental health does not satisfy users | 1.1. Very cold and distant services |
| | 1.2. Lack of medical record |
| 2. Problems with the schedules and timetable of attention | 2.1. Non-fixed schedules |
| | 2.2. Unfulfilled schedules |
| **Provider of the services** <br> 3. Training of health care professionals in telemental health services | 3.1. The rapport between patients and healthcare professionals |
| | 3.2. Lack of connectivity |
| 4. Need for integration into the health system | 4.1. Lack of infrastructure |
| | 4.2. Coordination with other ministries |

## Problems with the schedules and timetable of attention

The issues raised by users regarding the scheduling and timing of telemental health appointments shed light on the challenges associated with aligning virtual consultations with users' daily lives. One user highlighted the inconvenience of receiving calls during specific hours, such as eight to nine in the evening. This timing was noted as less than ideal, especially when users were engrossed in household activities, such as cooking. The unexpected nature of the calls created disruptions and surprises, leading to a less-than-optimal user experience.

> "But one hassle has been, for instance, getting a call around eight, eight-thirty, nine. It's not the best time, you know? Sometimes, you're already knee-deep in household chores, maybe cooking in the kitchen. It's a bit of a hassle because they ring you up, the phone goes off, you rush, caught up doing something urgent in the kitchen. And there it is, surprise, it's the doctor on the line."
>
> -D9: High-school teacher-

The second quote further emphasizes the inconsistency in adhering to scheduled appointments. Users expressed frustration when promised calls did not occur at the designated time or date, indicating a potential flaw in the reliability of telemental health scheduling systems. This lack of adherence to agreed-upon schedules not only disrupts users' routines but also raises concerns about the overall efficiency and effectiveness of telemental health services.

> "Of course, that's more or less what I was mentioning almost at the beginning, right? The issue where the doctor tells you they'll call you on a certain date at a certain time, but it doesn't happen at that time. Maybe it doesn't even happen on that date. So, it's like you expected the call, but then they don't call. It's a limitation I've noticed, a problem I've been able to identify."
>
> -D21: Healthcare worker- User-

These user experiences underscore the importance of flexible scheduling options in telemental health to accommodate users' diverse daily routines. Additionally, there is a need for improved communication and adherence to scheduled appointments to enhance user satisfaction and trust in this services. Addressing these challenges is crucial for optimizing the usability and acceptance of telemental health platforms.

## Training of healthcare professionals in telemental health services

The discourse surrounding the training of healthcare professionals in telemental health service brings attention to the varied experiences and viewpoints of practitioners adjusting to remote service provision. The first quote underscores the distinction between providing psychological support through telemental health and more formal evaluation or therapy. Despite the acknowledged challenge of building rapport over the phone, the practitioner emphasizes the practical advantages of telemental health, particularly in reaching a broader audience, including those in remote locations with limited economic means.

> "It's important to note that what we were providing was psychological support, not formal evaluation or therapy. Naturally, establishing rapport with individuals over the phone can be a bit more challenging. However, from a practical standpoint, I believe it's a more effective approach because it allows us to extend our services to a broader audience. We can

engage with people who live at a considerable distance, making it economically unfeasible for them to come in person. So, I see that as the primary advantage."

-D5: Telehealthcare worker-

Conversely, the second quote provides a firsthand account of the challenges encountered during teleconsultations. The mention of connectivity issues, interference, dropouts, and freezing problems reveals the technological barriers faced in the process. This personal experience sheds light on the complexities healthcare professionals confront when implementing telemental health, particularly in areas with limited infrastructure.

"I started in December 2021, and the experience wasn't very helpful, especially during teleconsultations or virtual sessions, as they call it. In the district where the health center is located, people lack easy access to networks and the internet. Most only have a cellphone connection, and when we managed to connect, there were often significant interference, dropouts, and freezing issues. We usually ended up resorting to a simple phone call. Many times, we started with the intention of using video to see the patient, but it typically turned into a phone call due to these connectivity challenges."

-D6: Telehealthcare worker-

Collectively, these quotes highlight the necessity for comprehensive training programs for healthcare professionals engaging in telemental health. Such training should not solely concentrate on the technical aspects of using telecommunication tools but also address the distinctive dynamics of remote patient interactions. Additionally, they underscore the significance of recognizing the limitations and hurdles associated with the implementation of telemental health services, with a focus on mitigating technological barriers to ensure effective and equitable healthcare delivery.

## Need for integration into the health system

In discussing the need for integration of telemental health into the health system, two key quotes provide insights into the challenges and requirements for a successful implementation.

"Not tied directly to regulations but rather to the resources needed for such interventions. It's crucial to have the know-how, and at that time, the ministry was gaining expertise. The telemental health department was still emerging, a limitation not just in mental health but overall. Implementation resources within services, like computerization, an electronic health record, and a reliable network, were lacking in the country. Users needed data, including mobile phones, for access. Especially for MINSA, serving 20 million people insured under the SIS, primarily due to limited economic resources. According to INEI reports, around 50-60% of people in some regions lacked a cellphone or computer in 2020-2021. You could have everything implemented, but if users lack network coverage and don't have the necessary equipment or data for connection, it becomes an important but insufficient tool for institutions in the Ministry of Health catering to a population with these economic characteristics."

-D3: Decision-maker-

Initiating with the first quote, it underscores the critical aspect of resources necessary for telemental health interventions. It highlights the evolving nature of the telemental health

department within the ministry, emphasizing the need for expertise. This issue is not limited to mental health but extends to the broader healthcare landscape. The lack of implementation resources, including computerization, electronic health records, and a reliable network, has been identified as a significant limitation. The quote emphasizes the importance of users having access to data and devices, particularly in economically constrained regions. The challenges faced by the Ministry of Health, serving a large population with limited resources, are highlighted through statistics indicating a lack of mobile phones or computers among a significant percentage of the population.

> "In 2020, updates to the National Mental Health Plan were made in the context of COVID, for instance. We reviewed these to align with specific regulations for our strategy implementation. Operational definitions were derived from MINSA documents. Additionally, there were other regulations, such as those related to database use. Managing databases with confidential information from users necessitates procedures, like registration overseen by the Ministry of Justice. Participation, determining focal points, and ensuring database protection are essential aspects beyond merely developing the service..."

> -D1: Decision-maker-

Conversely, the following quote delves into the regulatory and procedural aspects necessary for the integration of telemental health. It references updates in the National Mental Health Plan in response to the COVID context, signifying the dynamic nature of regulations. The operational definitions derived from MINSA documents illustrate the reliance on established guidelines. The mention of regulations related to database use highlights the importance of managing sensitive information. The necessity for procedures overseen by the Ministry of Justice, including registration and database protection, further emphasizes the regulatory framework and ethical considerations involved in telemental health implementation.

Collectively, these quotes underscore the multifaceted challenges in integrating telemental health into the health system. They emphasize the need for a comprehensive approach that addresses not only technical and resource-related challenges but also regulatory and ethical considerations to ensure a successful and ethical implementation within the broader healthcare ecosystem.

## Discussion

### Challenges in the implementation of telehealth in Peru

In Peru, as in other Latin American countries, the use of telehealth has not been widely extended throughout the country [28]. Initially, telehealth initiatives emerged from the need to provide healthcare services in rural areas. Before the pandemic, the legal framework for these services was limited to remote consultations between healthcare professionals. Later, a more appropriate legal framework was established, allowing broader patient access to telehealth services. Despite this, there is still a need to integrate the legal framework with health systems, especially in primary care settings [29]. In Latin America in general, the goal of telehealth is to address the shortage of healthcare personnel and specialized resources in medical care [30,31]. However, its main limitation is technological infrastructure. Other significant barriers include the type of healthcare provider and their care workflows, health and illness-related cultural factors, the healthcare system itself, and organizational investment [32].

The results of the interviews bring into discussion the perspectives on the limitations and barriers faced by both users and managers when planning and operating telehealth services in Peru. This study allows for an exploration of these viewpoints to establish a baseline that

will be useful for future developments in telehealth services in Peru. Among the first identified topics, opinions on user satisfaction with telehealth services were assessed. Many users expressed dissatisfaction with virtual care due to the lack of in-person interaction. However, other studies have shown that this perspective may improve when users have sufficient digital skills to understand the dynamics of online communication [33].

A major limitation to the widespread adoption of telehealth in Peru is limited Internet connectivity, which hinders the delivery of high-quality remote healthcare services [34]. These challenges can be addressed through significant improvements in infrastructure, including access to high-speed Internet. Despite this, many innovative asynchronous teleconsultation services have emerged in rural health centers, which is a promising development [29,35]. It is also crucial to develop digital skills among healthcare workers and patients. Digital literacy remains a gap that has not yet been considered in the implementation processes of digital health in Peru, making it necessary to establish strategies to improve it [36].

On the other hand, another key requirement for the efficient adoption of telehealth is the integration of information systems. Beyond the legal framework, this aspect requires a strong commitment from healthcare system leaders and authorities. A shift in perspective is needed—telehealth should not be seen solely as a tool to reduce access gaps or compensate for operational shortcomings. Instead, these new technologies can also provide healthcare systems with improvements in their commercial and operational indicators, as well as new data assets that benefit both patients and research through the proper use of digital clinical data [37].

Another aspect related to the differences between in-person and virtual care is the effectiveness of appointment scheduling. In Peru, this is a common issue for users, not only in telehealth services but even more frequently in in-person care, leading to high levels of dissatisfaction [38]. As an alternative, evidence shows that teleconsultations—whether by phone or video call—when properly managed with continuous support, can provide comparable clinical outcomes and improve patient satisfaction by eliminating the need for travel to healthcare facilities and reducing delays in addressing urgent and emergency cases [39]. Telehealth, whether via phone or video, has emerged as an effective alternative to in-person consultations, particularly in primary care and mental health services [40]. Video consultations, in particular, tend to deliver equal or even better clinical and economic outcomes than phone consultations, especially when visual information is essential for patient engagement [41]. However, evidence on the clinical benefits of telemedicine remains inconclusive, and further research is needed to establish its safety and cost-effectiveness [42].

## Holistic and user-centered paradigm for telemental health implementation

The study's outcomes highlight the essential need for a shift toward a holistic and user-centered approach to telemental health implementation. This necessitates a comprehensive reevaluation extending beyond the conventional focus on technological aspects to encompass the intricate interplay of human and systemic elements in the remote delivery of healthcare. The integration of such an approach has the potential to address multifaceted challenges, paving the way for the realization of telemental health's transformative force in ensuring accessible, equitable, and high-quality healthcare services [43].

A systematic review of the future of telemedicine [44]. Confirmed that a holistic approach to implementation and execution is needed to ensure that it incorporates technologies, organizational structures, change management of professionals and users, economic budget, social impact, perceptions and usability, policy and legislation. This should include new and existing theories on telemedicine that will help to keep the service in a constant state of change and improvement. By doing so, we can identify and discuss a wide range of barriers, opportunities,

and impacts related to telehealth implementation across our healthcare system [45]. Autors as Van D. et al. argued that there are various frameworks for evaluating factors that involve the implementation of a holistic telehealth model in health services. The Diffusion of Innovation Framework explains that technical, behavioral, economic, and organizational barriers must be explored. On the other hand, regarding Readiness Frameworks, the basic readiness that involves aspects of planning and integration, as well as technological, learning, social, and political elements, must be considered [44].

Furthermore, an analysis of technological infrastructure, connectivity, and interoperability is necessary to identify barriers to successful telehealth implementation. This will allow us to know what resources should be allocated for logistical implementation. Besides, we need to educate and train both healthcare professionals and patients. In interviews, it was noted that patients do not have the technical knowledge, which limits the adoption of telehealth. In the same way, many healthcare professionals do not have the necessary training to use the platforms. Finally, there is a need for trust and confidence that users will feel comfortable using these systems [46].

In this regard, the qualitative evaluation methodology using semi-structured interviews allows for an in-depth exploration of those aspects. Authors like T. Duran et al. refer to as user identification and contextual factors [47]. This lays the foundation for the application of co-design methodologies and technological solutions, such as Design Thinking, based on user needs [48].

In summary, adopting holistic and user-centered approach through qualitative evaluation methodologies to explore different aspects of the feasibility for to telemental health services holds promise in ensuring accessibility, equity, and quality, particularly in mental health care.

## Multifaceted challenges and transformative potential

The identified multifaceted challenges inherent in telemental health implementation serve as pivotal points for stakeholders to intervene. By strategically addressing these challenges, stakeholders can unlock the full potential of telemental health as a transformative force. Notably, experiences from multifaceted telehealth interventions indicate that this holistic approach is not only effective but can also serve as a viable solution for crafting intricate telehealth interventions, particularly in resource-constrained settings where time and resources are limited [49].

Based on the interviews, several challenges associated with the use of telemedicine emerged, particularly regarding its implementation. It is essential to highlight the interaction between users, professionals, technology, and the healthcare system. A primary challenge is related to the adoption of the necessary technology by both professionals and users. In this regard, it is crucial that both parties familiarize themselves with the technology. However, it is not only about familiarization; training activities (knowledge) and skill-building (ability to use) are fundamental [50,51]. Strengthening the interaction between professionals and users when utilizing these technologies is also vital, as the goal is to ensure that technology does not become a barrier to establishing a good rapport, which is critical in this type of service [52].

A second challenge relates to the operational aspects of telemedicine services. Familiarizing oneself with the service itself is distinct from merely understanding the technology [50,53,54]. The former involves practical knowledge within the specific service context. One aspect observed in the results, which should be taken into account, is the order and logic that should underpin the scheduling process for appointments. Even though the service does not occur in a physical space per se, it must maintain a structure and sequence [54].

Lastly, although not mentioned by the interviewees, other challenges related to the privacy and security of user information, as well as service access, should be considered [55]. Privacy and security issues stem from potential data breaches, making it essential to use platforms that ensure information security and comply with current privacy regulations [51,55]. Regarding access, it is necessary to consider the needs and capabilities of the users who will receive the services, including the possibility of analyzing strategies to guarantee access, such as subsidies [51].

### Policy emphasis and future directions

The current landscape of telehealth policy has predominantly focused on expanding access to specialized services and allowable originating sites [56]. Looking to the future, a crucial paradigm shift is advocated, urging policymakers to enhance telehealth equity through the implementation of system-level changes. These changes should consider the aspects referenced and discussed, such as the use of holistic paradigms for service implementation, technology adoption aspects, operational aspects of services, security and privacy issues, and even infrastructure-related aspects. The goal is to address and overcome the barriers that have historically impeded access for marginalized and low-income populations, thereby transforming the current telemental health ecosystem into a more inclusive and equitable one [57].

### Conclusion

In conclusion, the study's findings contribute substantively to the ongoing discourse on telemental health implementation. The emphasis on a holistic, user-centered approach aligns with the evolving needs of healthcare delivery in a remote context. The insights provided underscore the importance of not only advancing technological capabilities but also understanding and addressing the complex socio-cultural and systemic dynamics to fully unlock the potential of telehealth in improving healthcare accessibility and quality. By focusing on these multifaceted aspects, stakeholders can better navigate the challenges and opportunities that telehealth presents, ultimately transforming the healthcare landscape into one that is more inclusive, equitable, and effective.

### Supporting information

**S1 File. Semi-structured interview format.** Questionnaires used for the semi-structured interviews with external users, internal users, and decision-makers.
(DOCX)

**S2 File. Identified development requirements.** Software requirements and procedures identified in the study.
(DOCX)

**S3 File. Transcripts of the interviews.** Transcripts of the interviews by topic, in Spanish.
(DOCX)

**S4 File. Atlas-TI code file.** It contains the codes that have been created and used during the data coding process, including labels, categories, or themes by topic, in Spanish.
(ZIP)

### Author contributions

**Conceptualization:** John Astete Cornejo, Liliana Cruz-Ausejo, Jimmy Cainamarks Alejandro, Miguel Angel Burgos-Flores, Juan Ambrosio-Melgarejo, Jaime Rosales Rimache, Sebastián García Cruz.

**Data curation:** Liliana Cruz-Ausejo, Jimmy Cainamarks Alejandro, Miguel Angel Burgos-Flores, Juan Ambrosio-Melgarejo, Sebastián García Cruz.

**Formal analysis:** Liliana Cruz-Ausejo, Jimmy Cainamarks Alejandro, Miguel Angel Burgos-Flores, Juan Ambrosio-Melgarejo, Sebastián García Cruz.

**Funding acquisition:** John Astete Cornejo, Liliana Cruz-Ausejo, Jimmy Cainamarks Alejandro, Juan Ambrosio-Melgarejo.

**Investigation:** Liliana Cruz-Ausejo, Jimmy Cainamarks Alejandro, Miguel Angel Burgos-Flores, Juan Ambrosio-Melgarejo, Sebastián García Cruz.

**Methodology:** Liliana Cruz-Ausejo, Jimmy Cainamarks Alejandro, Miguel Angel Burgos-Flores, Juan Ambrosio-Melgarejo, Sebastián García Cruz.

**Resources:** Liliana Cruz-Ausejo, Jimmy Cainamarks Alejandro, Miguel Angel Burgos-Flores, Juan Ambrosio-Melgarejo, Jaime Rosales Rimache, Sebastián García Cruz.

**Software:** Jimmy Cainamarks Alejandro, Miguel Angel Burgos-Flores.

**Supervision:** John Astete Cornejo, Liliana Cruz-Ausejo, Miguel Angel Burgos-Flores, Juan Ambrosio-Melgarejo, Jaime Rosales Rimache, Sebastián García Cruz.

**Validation:** Liliana Cruz-Ausejo, Jimmy Cainamarks Alejandro, Miguel Angel Burgos-Flores, Juan Ambrosio-Melgarejo.

**Visualization:** Liliana Cruz-Ausejo, Miguel Angel Burgos-Flores, Juan Ambrosio-Melgarejo, Sebastián García Cruz.

**Writing – original draft:** Liliana Cruz-Ausejo, Jimmy Cainamarks Alejandro, Miguel Angel Burgos-Flores, Juan Ambrosio-Melgarejo, Jaime Rosales Rimache, Sebastián García Cruz.

**Writing – review & editing:** Liliana Cruz-Ausejo, Jimmy Cainamarks Alejandro, Miguel Angel Burgos-Flores, Juan Ambrosio-Melgarejo, Sebastián García Cruz.

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
