## [Decision Letter · Decision Letter 0]

14 May 2024

PONE-D-23-44037Barriers and limitations to the development of a tele-mental health service for workers - A user-centered approachPLOS ONE

Dear Dr. Cruz- Ausejo,

Thank you for submitting your manuscript to PLOS ONE. After careful consideration, we feel that it has merit but does not fully meet PLOS ONE’s publication criteria as it currently stands. Therefore, we invite you to submit a revised version of the manuscript that addresses the points raised during the review process.

Upon careful consideration of the reviewers' comments, it has become apparent that significant revisions are necessary before we can proceed towards acceptance. In particular, Reviewer 2 has raised a critical point regarding the lack of a theoretical framework underpinning your work. This is a fundamental aspect that needs to be addressed to strengthen the manuscript.

We look forward to receiving your revised manuscript.

Kind regards,

Josep Vidal-Alaball, MD, PdD, MPH

Academic Editor

PLOS ONE

Journal Requirements:

2. Thank you for stating the following financial disclosure: "Funding: This work was funded by the Agencia Española de Cooperación Internacional para el Desarrollo (AECID, in Spanish) by Programa Nacional de Investigación Científica y Estudios Avanzados- CONCYTEC- Perú, under contract: N° 076-2021-PROCIENCIA."  

Additional Editor Comments:

Dear Authors,

Thank you for submitting your manuscript to our journal. We have completed the review process and received valuable feedback from our reviewers.

Upon careful consideration of the reviewers' comments, it has become apparent that major significant revisions are necessary before we can proceed towards acceptance. In particular, Reviewer 2 has raised a critical point regarding the lack of a theoretical framework underpinning your work. This is a fundamental aspect that needs to be addressed to strengthen the manuscript.

Reviewers' comments:

Reviewer's Responses to Questions

**Comments to the Author**

1. Is the manuscript technically sound, and do the data support the conclusions?

Reviewer #1: Partly

Reviewer #2: Partly

2. Has the statistical analysis been performed appropriately and rigorously? 

Reviewer #1: N/A

Reviewer #2: N/A

3. Have the authors made all data underlying the findings in their manuscript fully available?

Reviewer #1: Yes

Reviewer #2: No

4. Is the manuscript presented in an intelligible fashion and written in standard English?

Reviewer #1: Yes

Reviewer #2: Yes

5. Review Comments to the Author

Reviewer #1: 1. Abstract:

Well written. However, the authors need to provide a further breakdown of the number of participants, i.e.15, 5 and three, so that the characteristics of the participants are clear. The last sentence of the introduction in the abstract could read better if simplified and shortened. It loses meaning in its current state.

2. Introduction

Clear and to the point. Please check line 78 if the percentage is well represented.

3. Data collection

Although the authors mention that they used semi-structured interviews, the following is not clear:

- What was the opening question? What were some of the probing questions used?

- At what point did the interviews stop or at what point was the data sufficient or what guided the authors to stop data collection at 23 with different types of participants (15,5 and 3)?

4. Discussion

While the implications and future directions are outlined clearly, the discussion is scanty. There is a need to broaden it as, in its current form, it is narrow-focused.

Reviewer #2: This study address an important topic: Which barriers and facilitators exist for telemental health services for vulnerable workers in countries were telecommunication may be limited or lacking. They conducted 23 semi-structured interviews with receivers (police officers and teachers) and planners/providers (healthcare personnel and administrators).

Here are my comments:

General comments: I think language should receive more attention, preferrably using an authorized translator.

I think the scientific basis is lacking, and there is no attempt to describe the theoretical basis on which the identification og barriers and facilitators were identified and sorted.

Full Title: Barriers and limitations to the development of a tele-mental health service for workers - A user-centered approach I think the findings in this study apply to Peru, which is also stated in the abstract, and the title should refer to Peru. Suggested change: Barriers and limitations to the development of a telemental health service for workers in Peru - A user-centered approach

Abstract

32 users and attendance The meaning of «attendance» here is unclear

33 mental telehealth Why not continue calling it telemental health

37 some of them were related to user dissatisfaction emphasizing the need for personalized solutions beyond technical aspects This could be specified more, what kind of dissatisfaction?

38 Scheduling issues call for flexibility and improved communication Meaning?

Introduction

General comment: I think the authors should define their use of the telemental health service term, and they should describe the plethora of teleservices that may be available

51 is disrupting global healthcare services Is disrupt the preferred verb? Consider using terms like challenging or altering. Disrupt definition: to prevent something, especially a system, process, or event, from continuing as usual or as expected

65 It is noteworthy, however, that the Ministry of Health faces certain limitations as the sole regulatory entity for telehealth in Peru Reference?

69 Consequently, the imperative to attend to these concerns remotely Meaning?

71 Notably, the global experience underscores that, despite the widespread activation of various e-screening initiatives in routine clinical practice throughout the COVID 19 era, comprehensive studies evaluating their appropriateness, efficiency, and efficacy remain notably scarce Avoid using notably twice in one sentence.

MATERIAL AND METHODS

91 Spanning durations of 10 to 35 minutes, the interviews were conducted to comprehend perceptions, needs, barriers, and 99 facilitators related to the implementation of mental telehealth services On what theoretical ground were barriers and facilitators identified? Who conducted the interviews – what professional background did they have? Did you recruit interviewees from urban or rural areas, or both?

117 Decision makers How where these recruited?

125 Upon orally expressing their agreement to participate, audiovisual or voice recording, as applicable, was initiated. A Did you not receive written consent?

133 . Adopting a phenomenological approach, the codes have been tailored to align with the nuances apparent in our dataset Meaning?

134 This dynamic process has enhanced our coding framework, directing our focus towards the four key themes associated with the mental health telecare and screening service. The primary themes and subthemes that emerged from this process are detailed in Table 1 Wouldn’t this be more appropriately put in results section, including table?

Results

General comment: From 23 interviews 8 quotes are presented, for instance the reference to «healthcare worker» does not clarify whether it is quotes from the same healthcare worker or various (that is: it is impossible for the reader to assess it). The same problem arises for «Decision-maker». One could extend the characterization of each responder, by indicating if rural or urban, if not in conflict with anonymity.

General comment: The healthcare workers represent a completely different position as compared to the other interviewees, since these represent the service and not the receiver of services. Thus, I find it problematic that their experiences are included in the same table. See suggestion above.

137 Table 1 1.2. Lack of medical record Where this issue addressed? If not, consider omitting it from the table

137 Table 1 Consider revising the table, indicating that item 1 and 2 represent «Receiver of the services» and 3 and 4 represent «Provider/planner of the services»

Discussion

General comments: In discussion there is a chance to discuss strengths and limitations of the study. They should also compare their own findings with existing literature, and they should make an attempt to guide clinicians and decision makers, base don theri findings. In addition, the discussion addresses to a very limited degree the challenges that were identified in the results section. Rather than describing approaches to improve telehealthcare in Peru in very general terms, they should use this section to develop more specific thoughts on the significance of their findings

270 The study's outcomes underscore the imperative shift toward a holistic and user-centered approach to 271 telehealth implementation How?

6. PLOS authors have the option to publish the peer review history of their article (what does this mean? ). If published, this will include your full peer review and any attached files.

**Do you want your identity to be public for this peer review?** For information about this choice, including consent withdrawal, please see our Privacy Policy .

Reviewer #1: **Yes: ** Andile Glodin Mokoena-de Beer

Reviewer #2: **Yes: ** Eivind Aakhus

---

## [Author Response · Author response to Decision Letter 1]

7 Jan 2025

Dear Editor,

Thank you for the opportunity to review the manuscript entitled “Barriers and limitations to the development of a tele-mental health service for workers- A user-centered approach”.

In the following paragraphs, we proceed to comment on the changes made according to each observation made by the reviewers and Academic Editor.

Academic Editor Comments:

1. Please ensure that your manuscript meets PLOS ONE's style requirements, including those for file naming. The PLOS

ONE style templates can be found at

Answer: All the changes suggested by the reviewers have been made.

2. Thank you for stating the following financial disclosure: "Funding: This work was funded by the Agencia Española de

Cooperación Internacional para el Desarrollo (AECID, in Spanish) by Programa Nacional de Investigación Científica y Estudios Avanzados- CONCYTEC- Perú, under contract: N° 076-2021-PROCIENCIA."

Please state what role the funders took in the study. If the funders had no role, please state: ""The funders had no role in study design, data collection and analysis, decision to publish, or preparation of the manuscript."" If this statement is not correct you must amend it as needed.

Answer: The statement of non-involvement by the funders in the study was included.

Answer: There are no restrictions on sharing the anonymized data obtained in the study. Accordingly, the full transcripts of the interviews conducted with the participants, along with the Atlas.TI code file, are attached as a Supporting information file.

Reviewer #1:

The manuscript reflects a justifiable need for this study to be conducted and dissemination of data thereof. The methodology used is appropriate to elicit information from the participants. However, the discussion is scanty and does not clear overview of the findings against existing studies in other countries. As such, this manuscript requires minor revisions to give it more meaning at a standard that could be published by the journal.

Answer: The discussion topics in the manuscript were expanded according to subtopics. The discussion was expanded to consider different international contexts.

Abstract:

Well written. However, the authors need to provide a further breakdown of the number of participants, i.e.15, 5 and three, so that the characteristics of the participants are clear. The last sentence of the introduction in the abstract could read better if simplified and shortened. It loses meaning in its current state.

Answer: The observation has been acknowledged and modified in the abstract.

Introduction

Clear and to the point. Please check line 78 if the percentage is well represented.

Answer: The referred percentage was the one reported by the government in the official gazette.

Data collection

Although the authors mention that they used semi-structured interviews, the following is not clear:

What was the opening question? What were some of the probing questions used?

At what point did the interviews stop or at what point was the data sufficient or what guided the authors to stop data collection at 23 with different types of participants (15,5 and 3)?

Answer: The base questions of the semi-structured interview, as approved in the research protocol, are attached as a Supporting information file. These questions introduced the topic, and as can be observed in the interview transcripts, they allowed the interviewer to ask follow-up questions and explore certain criteria in greater depth. The interviews ceased upon reaching the sample approved by the ethics and research committee, as the three groups contribute to the identification of development requirements for the same software.

Discussion

While the implications and future directions are outlined clearly, the discussion is scanty. There is a need to broaden it as, in its current form, it is narrow-focused.

Answer: The discussion was extended.

Reviewer #2:

This study address an important topic: Which barriers and facilitators exist for telemental health services for vulnerable workers in countries were telecommunication may be limited or lacking. They conducted 23 semi-structured interviews with receivers (police officers and teachers) and planners/providers (healthcare personnel and administrators).

General comments: I think language should receive more attention, preferrably using an authorized translator. I think the scientific basis is lacking, and there is no attempt to describe the theoretical basis on which the identification of barriers and facilitators were identified and sorted.

Answer: Thank you very much for your feedback. We strive to take all of your observations into account when making the necessary corrections. Grammar corrections were also made to improve comprehension.

Full Title: Barriers and limitations to the development of a tele-mental health service for workers - A user-centered approach I think the findings in this study apply to Peru, which is also stated in the abstract, and the title should refer to Peru. Suggested change: Barriers and limitations to the development of a telemental health service for workers in Peru - A user-centered approach

Answer: The observation has been resolved as “Barriers and limitations to the development of a telemental health service for workers in Peru - A user-centered approach.”

Abstract

32 users and attendance The meaning of «attendance» here is unclear

33 mental telehealth Why not continue calling it telemental health

37 some of them were related to user dissatisfaction emphasizing the need for personalized solutions beyond technical aspects This could be specified more, what kind of dissatisfaction?

38 Scheduling issues call for flexibility and improved communication Meaning?

Answer: Text related to “attendance” was replacing by “providers of mental telehealth”. The user's dissatisfaction centered on the lack of customizable solutions, like fast communication. The grammar was revised to clarify this.

Introduction

General comment: I think the authors should define their use of the telemental health service term, and they should describe the plethora of teleservices that may be available

51 is disrupting global healthcare services Is disrupt the preferred verb? Consider using terms like challenging or altering.

Answer: The definition and services related to telemental health were included, as well as the correction in the term "disrupting."

Disrupt definition: to prevent something, especially a system, process, or event, from continuing as usual or as expected

65 It is noteworthy, however, that the Ministry of Health faces certain limitations as the sole regulatory entity for telehealth in Peru Reference?

69 Consequently, the imperative to attend to these concerns remotely Meaning?

71 Notably, the global experience underscores that, despite the widespread activation of various e-screening initiatives in routine clinical practice throughout the COVID 19 era, comprehensive studies evaluating their appropriateness, efficiency, and efficacy remain notably scarce Avoid using notably twice in one sentence.

Answer: The recommendations were adopted, including the references and improving the writing according to the suggestions.

MATERIAL AND METHODS

91 Spanning durations of 10 to 35 minutes, the interviews were conducted to comprehend perceptions, needs, barriers, and 99 facilitators related to the implementation of mental telehealth services On what theoretical ground were barriers and facilitators identified? Who conducted the interviews – what professional background did they have? Did you recruit interviewees from urban or rural areas, or both?

Answer: The barriers and facilitators were identified using a holistic and user-centered approach. The interviews were conducted by the study's researchers, all of whom were trained in qualitative research methodologies. Additionally, all interviewees were recruited from urban areas. These aspects were incorporated into the writing of Material and Methods.

117 Decision makers How where these recruited?

Answer: The indicated precision was included in the writing.

125 Upon orally expressing their agreement to participate, audiovisual or voice recording, as applicable, was initiated. A Did you not receive written consent?

Answer:The prior application of consent is described in the section "Ethical Statement and Informed Consent."

133 . Adopting a phenomenological approach, the codes have been tailored to align with the nuances apparent in our dataset Meaning?

Answer: The barriers and facilitators were identified using a holistic and user-centered approach.

134 This dynamic process has enhanced our coding framework, directing our focus towards the four key themes associated with the mental health telecare and screening service. The primary themes and subthemes that emerged from this process are detailed in Table 1 Wouldn’t this be more appropriately put in results section, including table?

Answer: Table 1 was moved to the Results section.

Results

General comment: From 23 interviews 8 quotes are presented, for instance the reference to «healthcare worker» does not clarify whether it is quotes from the same healthcare worker or various (that is: it is impossible for the reader to assess it).

The same problem arises for «Decision-maker». One could extend the characterization of each responder, by indicating if rural or urban, if not in conflict with anonymity.

Answer: Modifications were made to the presentation of the interviewees by assigning them alfanumerical identifiers (D#) to distinguish between them.

General comment: The healthcare workers represent a completely different position as compared to the other interviewees, since these represent the service and not the receiver of services. Thus, I find it problematic that their experiences are included in the same table. See suggestion above.

137 Table 1 1.2. Lack of medical record Where this issue addressed? If not, consider omitting it from the table

137 Table 1 Consider revising the table, indicating that item 1 and 2 represent «Receiver of the services» and 3 and 4 represent «Provider/planner of the services»

Answer: The changes were made according to the suggestions.

Discussion

General comments: In discussion there is a chance to discuss strengths and limitations of the study. They should also compare their own findings with existing literature, and they should make an attempt to guide clinicians and decision makers, base don theri findings. In addition, the discussion addresses to a very limited degree the challenges that were identified in the results section. Rather than describing approaches to improve telehealthcare in Peru in very general terms, they should use this section to develop more specific thoughts on the significance of their findings

270 The study's outcomes underscore the imperative shift toward a holistic and user-centered approach to 271 telehealth implementation How?

Answer: More details were added based on feedback.

The authors consider that we have provided a response to all the observations made to the manuscript; Therefore, we send this document for your review.

Kind regards

---

## [Editor Report · Decision Letter 1]

24 Jan 2025

PONE-D-23-44037R1Barriers and limitations to the development of a telemental health service for workers in Peru- A user-centered approachPLOS ONE

Dear Dr. Cruz- Ausejo,

Thank you for submitting your manuscript to PLOS ONE. After careful consideration, we feel that it has merit but does not fully meet PLOS ONE’s publication criteria as it currently stands. Therefore, we invite you to submit a revised version of the manuscript that addresses the points raised during the review process.

We look forward to receiving your revised manuscript.

Kind regards,

Josep Vidal-Alaball, MD, PdD, MPH

Academic Editor

PLOS ONE

Additional Editor Comments (if provided):

Dear Authors,

PLOS ONE considers qualitative and mixed-methods studies for publication. We recommend that authors use the COREQ checklist, or other relevant checklists listed by the Equator Network, such as the SRQR, to ensure complete reporting (http://journals.plos.org/plosone/s/submission-guidelines#loc-qualitative-research). In general, we would expect qualitative studies to include the following: 1) defined objectives or research questions; 2) description of the sampling strategy, including rationale for the recruitment method, participant inclusion/exclusion criteria and the number of participants recruited; 3) detailed reporting of the data collection procedures; 4) data analysis procedures described in sufficient detail to enable replication; 5) a discussion of potential sources of bias; and 6) a discussion of limitations.

---

## [Author Response · Author response to Decision Letter 2]

3 Mar 2025

Thank you for the opportunity to review the manuscript entitled “Barriers and limitations to the development of a tele-mental health service for workers- A user-centered approach”. The following section addresses the resolution of the received comments:

1. Abstract:

Well written. However, the authors need to provide a further breakdown of the number of participants, i.e.15, 5 and three, so that the characteristics of the participants are clear. The last sentence of the introduction in the abstract could read better if simplified and shortened. It loses meaning in its current state.

Response: The observations had already been resolved in the last version sent

2. Introduction

Clear and to the point. Please check line 78 if the percentage is well represented.

Response: Although this is the percentage figure provided by an official source, the wording has been modified according to the observation.

3. Data collection

Although the authors mention that they used semi-structured interviews, the following is not clear:

- What was the opening question? What were some of the probing questions used?

- At what point did the interviews stop or at what point was the data sufficient or what guided the authors to stop data collection at 23 with different types of participants (15,5 and 3)?

Response: The interview began with introductory questions designed to explore general perceptions of telehealth in mental health. Subsequently, probing questions were used to delve deeper into participants' experiences and their perspectives on the care process. Additionally, the Supporting Information section of the manuscript includes S1 File: Semi-Structured Interview Format, which contains the interview guide used in this study.

The interviews were conducted until data saturation was reached, meaning no new themes or additional relevant information emerged. The sample included three decision-makers, who represented the entirety of the actors responsible for the regulation and implementation of telehealth at that time, a nascent service in the Peruvian context. Although three distinct groups participated (users, providers, and decision-makers), their contributions were integrated into the formulation of a single telemental health service, confirming that data saturation had been achieved.

4. Discussion

While the implications and future directions are outlined clearly, the discussion is scanty. There is a need to broaden it as, in its current form, it is narrow-focused.

Response: More discussion elements were incorporated into the manuscript.

Additional Editor Comments (if provided):

Dear Authors,

PLOS ONE considers qualitative and mixed-methods studies for publication. We recommend that authors use the COREQ checklist, or other relevant checklists listed by the Equator Network, such as the SRQR, to ensure complete reporting (http://journals.plos.org/plosone/s/submission-guidelines#loc-qualitative-research). In general, we would expect qualitative studies to include the following: 1) defined objectives or research questions; 2) description of the sampling strategy, including rationale for the recruitment method, participant inclusion/exclusion criteria and the number of participants recruited; 3) detailed reporting of the data collection procedures; 4) data analysis procedures described in sufficient detail to enable replication; 5) a discussion of potential sources of bias; and 6) a discussion of limitations.

Response: We applied the COREQ guidelines for reporting qualitative research in this manuscript. A copy of the checklist is attached with the submission. Furthermore, we believe that the editorial requirements have been met.

We have addressed all submitted observations and believe the manuscript is now ready for further review and to proceed with the publication process.

Best regards,

The Authors.

---

## [Editor Report · Decision Letter 2]

5 Mar 2025

Barriers and limitations to the development of a telemental health service for workers in Peru- A user-centered approach

PONE-D-23-44037R2

Dear Dr. Cruz- Ausejo,

We’re pleased to inform you that your manuscript has been judged scientifically suitable for publication and will be formally accepted for publication once it meets all outstanding technical requirements.

Kind regards,

Josep Vidal-Alaball, MD, PdD, MPH

Academic Editor

PLOS ONE
---

## [Editor Report · Acceptance letter]

PONE-D-23-44037R2

PLOS ONE

Dear Dr. Cruz-Ausejo,

I'm pleased to inform you that your manuscript has been deemed suitable for publication in PLOS ONE. Congratulations! Your manuscript is now being handed over to our production team.

Kind regards,

on behalf of

Dr. Josep Vidal-Alaball

Academic Editor

PLOS ONE